# Defining the Recipe for an Optimal Rotavirus Vaccine Introduction in a High-Income Country in Europe

**DOI:** 10.3390/v14020425

**Published:** 2022-02-18

**Authors:** Baudouin Standaert, Bernd Benninghoff

**Affiliations:** 1HEBO bv, 2020 Antwerpen, Belgium; 2Research Group Care and Ethics, Faculty of Medicine and Life Sciences, University of Hasselt, 3500 Hasselt, Belgium; 3GSK, 1300 Wavre, Belgium; bernd.benninghoff@gmx.de

**Keywords:** rotavirus vaccination, implementation, optimisation, high income country, long-term effect

## Abstract

Observational data over 15 years of rotavirus vaccine introduction in Belgium have indicated that rotavirus hospitalisations in children aged <5 years plateaued at a higher level than expected, and was followed by biennial disease peaks. The research objective was to identify factors influencing these real-world vaccine impact data. We constructed mathematical models simulating rotavirus-related hospitalisations by age group and year for those children. Two periods were defined using different model constructs. First, the vaccine uptake period encompassed the years required to cover the whole at-risk population. Second, the post-uptake period covered the years in which a new infection/disease equilibrium was reached. The models were fitted to the observational data using optimisation programmes with regression and differential equations. Modifying parameter values identified factors affecting the pattern of hospitalisations. Results indicated that starting vaccination well before the peak disease season in the first year and rapidly achieving high coverage was critical in maximising early herd effect and minimising secondary sources of infection. This, in turn, would maximise the reduction in hospitalisations and minimise the size and frequency of subsequent disease peaks. The analysis and results identified key elements to consider for countries initiating an optimal rotavirus vaccine launch programme.

## 1. Introduction

Development and implementation of an appropriate vaccination programme with high vaccine coverage should help to control transmissible infections in a population [1]. Vaccination should reduce infection spread, with the possibility of disease elimination and even disease eradication [2] as has occurred previously with vaccines against childhood infections such as smallpox, polio, measles, varicella, and others [3]. However, vaccine success is not always guaranteed because infections and their transmission vary. Moreover, not all developed vaccines will stop infection spread immediately and sustainably. For example, the flu vaccine must be adjusted each year and there is still uncertainty over who should be vaccinated to achieve the best results for disease control [4].

Newer vaccines developed against diseases caused by rotavirus, *Streptococcus pneumoniae*, human papillomavirus or meningococci were also subject to similar issues around optimum implementation to reach maximum disease control benefit as quickly as possible [5,6]. Recent findings indicated that rotavirus vaccine implementation has resulted in different outcomes depending on the initiation of its launch [7]. When the new rotavirus vaccines first came onto the market in high-income countries (HICs) in 2006, there was confidence that these vaccines, with their mode of action generating an ‘infection-like’ reactivity that does not lead to severe health outcomes, would move to disease elimination [8,9], as predicted by the first published dynamic models based on results from testing the vaccine in Phase III studies in HICs [10,11]. However, these promising modelling outcomes were questioned when the first results on medium-term effects of the vaccines were reported [12,13,14]. Universal mass vaccination (UMV) campaigns in HICs were initiated in the United States of America (USA), Belgium, and Austria in 2006, but the results of those programmes differed from the predicted modelling results after a few years of vaccine implementation [15,16,17]. To explain the causes of the discrepancies between the real-world data on rotavirus vaccination compared with the early model predictions, we constructed a mathematical model for the early years after vaccine introduction including variables that could affect the outcome [18], based on observed data from the country with the longest detailed observations of rotavirus vaccination impact over time (the RotaBIS study in Belgium) [19]. In the analysis presented here we have extended this research by developing a second mathematical model covering later years. The analysis also discusses data from other HICs that initiated a UMV campaign such as Finland, United Kingdom (UK), Austria, Australia, and the USA, for comparison with the analysis findings. These countries were selected based on access to adequate published data [17,20,21,22,23,24]. The different strategies and experiences in different countries indicate that it was not obvious that the impact of rotavirus vaccination would be affected by the details of the vaccine programme launch. There is a need to understand the factors driving the real-world impact of rotavirus vaccination to identify an optimal strategy for vaccine introduction.

This research should help HICs that have not yet introduced rotavirus vaccination but now wish to begin a vaccination campaign such as Switzerland, France, the Netherlands, or Denmark [25] to identify an optimal vaccine implementation pathway. In principle, our recommendations could also be applicable to vaccination against other infections that may follow similar exposure routes and vaccination effects.

## 2. Materials and Methods

### 2.1. Key Features of Rotavirus Disease and the Vaccine

To understand the potential impact of vaccination over time, some background on rotavirus disease and the vaccine is needed. Rotavirus disease is mainly seen in very young children aged less than two years, often accounting for close to 80% of all the child rotavirus hospital disease events in the at-risk population aged under five years. The hospital disease age spread followed a Weibull distribution up to the age of 60 months (Figure 1 (RotaBIS data)) [26].

Rotavirus disease is seasonal in temperate climate conditions with a peak each year at the end of winter, from February to late April in the Northern Hemisphere [27]. The primary sources of infection in the at-risk population are infants aged between three and 14 months because that age group is most susceptible [28]. Older children in the at-risk population can also spread the virus and act as secondary infection sources. In the pre-vaccination period, the importance of this secondary infection source is limited due to the dominance of the younger age group [19].

There are two commercially available oral rotavirus vaccines commonly used in HICs, a 2-dose live attenuated vaccine derived from a single human strain (Rotarix^®^, GSK, Wavre, Belgium) and a 3-dose live attenuated human-bovine assorted vaccine with five strains (RotaTeq^®^, Merck Vaccines, Kenilworth, NJ, USA) [25]. The vaccines have a limited age indication for administration. Vaccination should be given prior to six months of age for the 2-dose vaccine, and prior to eight months of age for the 3-dose vaccine, because of the risk for a severe adverse event caused by the vaccine, intussusception, which increases with age [29]. As a result, when the vaccine is introduced, it cannot be administered as a catch-up vaccine to the whole at-risk population at once. Coverage needs to be built up gradually in the child population as successive cohorts of infants are vaccinated when eligible. The seasonality of rotavirus disease requires the selection of a starting point for the vaccine programme that can reach high accumulated vaccine coverage before the next disease peak season by maximising vaccine administration to all infants born from the start of the vaccination to the first disease peak season. High vaccine coverage in the peak season is crucial, because this is the time of highest virus transmission within the at-risk group, and this should be reduced by the vaccination [19].

The level of vaccine coverage during the first peak season after introduction has additional consequences for the level of indirect (also called herd) effect of the vaccination. Indirect effects can lead to important added protection of non-vaccinated at-risk children. Once the vaccine coverage has reached the 5-year implementation in the at-risk population, a new situation of infection spread and disease manifestation appears, depending on the initial coverage achieved. This is no longer a situation of coverage build-up, but a post-vaccine uptake period. A new state of infection equilibrium is achieved between susceptible, infected/infectious, and recovered children, exposed to a much lower rate of infection risk because of the vaccination [30].

### 2.2. Data Sources: The RotaBIS Study

RotaBIS is a database study initiated in 2007 and conducted in 11 hospitals in Belgium (eight general hospitals with a paediatric ward, three paediatric hospitals). The centres had 546 paediatric beds, representing 30.6% of a total of 1793 paediatric beds in Belgium [13]. All children aged ≤5 years were eligible for inclusion when hospitalised for diarrhoea if they had a rotavirus detection test performed at one of the participating centres. From 2005 until the end of 2006, enrolled children were considered the pre-vaccination study group and those enrolled from 2007 until the end of 2019 were the post-vaccination study group. The following information was recorded for each sample: patient’s birth date and gender; sample date; rotavirus test result; and date of admission and discharge. Hospitalisation was classified as AGE-driven if the stool sample was collected within 48 h of hospitalisation. The mean length of stay and total number of hospitalisation days were calculated for hospitalised patients. Rotavirus infections were considered community-acquired if a stool sample taken within 48 h of hospital admission was rotavirus-positive.

We compared the absolute numbers of rotavirus-positive test results between the pre-vaccination and post-vaccination study seasons with the number of positive tests in the pre-vaccination period as the reference. The underlying assumption is that the coverage area for each of the hospitals participating in the study remained the same across the whole study period. Therefore, the most relevant value for the comparison of pre-and post-vaccination is the average absolute number of positive tests observed per time unit. Vaccine sales data in Belgium showed that vaccine coverage was low before reimbursement. We therefore assumed that no children were vaccinated prior to reimbursement in November 2006.

Ethical approval was obtained annually from 2007 when the study was initiated and from 2018 for a 3-year contract period instead of annual data. Figure 2 summarises the observed RotaBIS data over time. It shows a rapid initial fall in hospitalisations, which levelled off at around year 3. This was followed by a pattern of biennial hospitalisation peaks starting in year 9 (Figure 2). We designated these two periods as the vaccine uptake period and the post-uptake period, respectively, with the division set at eight years after vaccine introduction, as indicated in Figure 2. Figure 2 also plots the results of two mathematical regressions, the first showing what would be expected with the vaccine effect fixed at 90%, coverage fixed at 90% and no other factors involved (‘Modelled fixed’), and the second showing the effect of adding indirect vaccine effect at 85% protection of unvaccinated children ‘(Ideal’). It can be seen that the observed RotaBIS data differed markedly from either of these regressions. The objective of our analysis was to investigate the reasons for these differences. We approached this by developing models to replicate the observed data, and then modified the parameters in the models to identify the key factors affecting the outcomes.

Combining these two different periods into a single model construct is challenging. The vaccine impact has a different dynamic, and as a consequence, the infection spread is also different in each period. We therefore modelled the vaccine uptake period and post-uptake periods separately, using different model types for each period. The vaccine uptake period used a linear regression equation with different parameters affecting the hospitalisation rate per unit time, and is described in detail elsewhere [18]. The present analysis added a second model for the post-uptake period, using a mathematical time-differential, compartmental model that replicated the hospitalisation peaks over time resulting from the impact of the vaccine on infection transmission dynamics.

### 2.3. Hypotheses to Test

#### 2.3.1. Hypothesis for the Vaccine Uptake Period

The null hypothesis (*H*_0_) was that the effect of vaccination observed during the vaccine uptake period resulted from a process that includes vaccine waning, which may reduce the vaccination effect, as often claimed in the literature [7,9,31]. The alternative hypothesis (*H_a_*) is that the vaccine impact seen is mainly the result of a combination of direct vaccine effect, indirect herd effect, and secondary sources of infection, with vaccine waning playing a much smaller role.

#### 2.3.2. Hypothesis for the Post-Uptake Period

Expectations for changes in hospitalisation over time with a vaccine of high efficacy and coverage are a continuous, proportional decline across the at-risk age groups, resulting in smaller hospitalisation peaks and time intervals between the peaks [32]. The first dynamic models published when rotavirus vaccine was introduced in HICs in 2006 indicated such a pattern [8]. The null hypothesis *H*_0_ is therefore that once a new infection equilibrium is reached after introducing the vaccine, hospitalisations would decline, leading to local elimination over time. The alternative hypothesis (*H_a_*) is that there is no further decline to be observed early in the post-uptake period, but that regular peaks may appear with smaller height and lower frequency than in the pre-vaccination period if the vaccine coverage at the start was not optimal. This pattern results from the influence of secondary sources of infection, and should be characterised by a change in the age distribution of hospitalised cases among the unvaccinated children in the post-uptake peaks compared with pre-vaccination.

### 2.4. Replicating the Observations through Modelling

#### 2.4.1. Modelling the Vaccine Uptake Period

To model the vaccine uptake period, we used a regression equation with known input variables with values estimated based on the RotaBIS data (Table 1), in which the dependent variable was the hospitalisation number. The model is described in detail elsewhere [18], and the full equation is defined as follows:



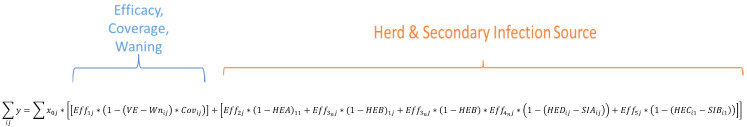





i=year



j=age group



y=rotavirus hospitalisations post−vaccination



$x_{0j}=rotavirus\;hospitalisations\;pre-vaccination$



Other codes are explained in Table 1.

As shown, the equation had two groups of independent variables affecting the outcome. The first group was related to the vaccination programme. Some variables had a positive direct impact on hospitalisations (vaccine efficacy and coverage), but others had a negative direct impact by reducing the vaccine effect (waning). The second group was related to the indirect impact of the vaccine, again, with positive effects in reducing the hospitalisations in unvaccinated children (the herd effect) and negative effects (presence of secondary sources of infection that cannibalise the herd effect). The value estimates of the variables were obtained from the RotaBIS study examining the impact of vaccination on hospitalisation in Belgium since the vaccine introduction in late 2006 [13,16,33,34,35] (Table 1).

#### 2.4.2. Modelling the Post-Uptake Period

The post-uptake period, as above-mentioned, is characterised by a new dynamic infection equilibrium in the at-risk population that may show hospitalisation peaks, as seen in the pre-vaccination period but at a much reduced height and frequency. The model must therefore simulate the observed hospitalisation peaks and be able to demonstrate that marginal changes in vaccine coverage affect the hospitalisation numbers. We selected a dynamic Susceptible–Infectious (SI) model, constructed using the Hamer–Soper (H–S) model type in which model predictions could remain stable over time [32,36]. This model allows us to identify the variables that produce the hospitalisation peaks at the observed frequency. However, with the H–S model, the list of variables is limited. A key variable was the peak hospitalisation number reached in the post-uptake peaks. We analysed the observed data of the RotaBIS study by week during the post-uptake period to define this peak value, which was around 50 hospitalisations per week in the at-risk population (Figure 3) [19].

A H–S model assumes constant replenishment of susceptible individuals via new-borns. The rate of new events (=hospitalisations) is jointly proportional to the number of susceptibles and the number of infectious individuals. The number of recoveries was also proportional to the number of events. Those not recovering were assumed to be equally contagious over the duration of an event. Those recovered were assumed to be immune and no longer contributed to the disease spread. The model design is presented in Figure 4, together with the two time-differential equations. Values had to be defined for the growth rate of susceptibles per time unit (α), the force of infection (λ), and the rate of recovery (γ). However, the latter is not critical because recovered individuals had no further impact on the disease spread.

The H–S equations were:$$x_{k+d}=x_{k}+\alpha -y_{k+1}$$
$$y_{k+d}=\lambda x_{k}y_{k}-\gamma y_{k}$$

The data input for the second model (Table 2) was also extracted from the RotaBIS database, complemented with an assumed value for the force of infection that replicated the dynamic process.

### 2.5. Analysis

The overall aim was to obtain three output results with the two models developed. The first one was to identify datasets for each model that achieved a best fit with the observed data for each period. This should help to identify the key factors affecting the results. A second aim was to evaluate the effect of improving vaccine coverage by 10 percentage points during the post-uptake-period, in a situation resembling the RotaBIS study data. The third area to explore was the effect on hospitalisations in the post-uptake period of improving vaccine coverage at the start of the first year in the vaccine uptake period to an ideal situation.

#### 2.5.1. Analysis of the Vaccine Uptake Period

To fit the model-generated data as closely as possible to the observed hospitalisation data at each annual time point, we used constrained optimisation, in which the objective function was to minimise the difference in hospitalisations between the observed and modelled data accumulated over the first eight years of the vaccine introduction. A best-fit software programme in Microsoft Excel, called Solver, was used to identify the value for each variable selected to produce the optimal match with the observational data while complying with specific constraints. These constraints were introduced at two levels. The first level specified the minimum and maximum range within which the values of the variables may vary in the regression equation (the four direct vaccine effect variables). The second level was related to the annual time- and age-specific fit of the data (the four indirect vaccine effect variables). Table 3 shows a matrix of age groups and annual observations, with eight areas having a common vaccine and/or disease effect during the vaccine uptake period. The colour code for each area will help to explain the sensitivity analysis conducted for each area. Table A1 in Appendix A provides more details of the optimisation analysis performed.

Scenario analysis was conducted to explore the effect on outcomes of vaccine coverage achieved in the first year. We introduced for that effect, a linear relationship between the herd effect and first-year vaccine coverage, in which a one percentage point rise in vaccine coverage in the first year resulted in a 1.64 percentage point increase in herd effect. This was estimated based on RotaBIS data. Vaccine coverage achieved during the peak season in the first year was around 52% and the maximum coverage that could be obtained by starting the vaccination earlier was 85%, while the herd effect measured in the first year was 31% and in an ideal scenario could reach 85%. A second linear relationship was introduced between this change in herd effect and the effect of secondary sources of infection in subsequent years in the same unvaccinated age groups. For each percentage point increase in herd effect, the reduction in the cannibalising effect of the secondary infection sources was 1.06 percentage points. The cannibalising effect of secondary sources of infection on herd effect in RotaBIS was estimated at 35% in the second year of observation, reaching 0% when the maximum herd effect was attained [37].

Sensitivity analysis evaluated four conditions separately: no herd effect; no secondary sources of infection; no waning present in the regression analysis; and a simulation of the compensation needed in waning to substitute the effect of removing secondary sources of infection to reach the observed hospitalisation result after eight years. We called this simulation the adjusted waning process.

#### 2.5.2. Analysis of the Post-Uptake Period

The first analysis evaluated the changes in age distribution in the hospitalisations between the pre-vaccination period and the peaks in the post-uptake period. We split hospitalisations into 13 age-groups and compared the distribution of hospitalisations between the pre-vaccination period and the first peak in the post-uptake period.

To run the dynamic model, data on estimated susceptibles at year 9 post-vaccination, when the stabilising dynamic infectious disease processes of the post-uptake period were operational, were needed. These were not available in the RotaBIS dataset, and were estimated using the following calculations. The pre-vaccination period in the RotaBIS study had around 1385 rotavirus hospitalisations per year in the child population up to five years old. This number represented about 2.5% of the at-risk child population, and therefore, a stable annual population number of 55,000 children was estimated as the input data for the population aged up to five years [26]. Each year, around 11,000 new children entered the group (55,000 divided by 5) and the same number left when reaching the age of five years. At year 8 post-vaccination, the newborn population had a vaccine coverage rate that reached 85%. In the remaining population aged up to one year, there was a slightly lower coverage of 83%. Therefore, in the group aged 0–1 years at year 9, the susceptible group was estimated at 1760 children, and in the total population aged up to five years, the average number of unvaccinated children was around 10,340 or 19% of the total at-risk child population. The number of susceptibles needed to induce a hospitalisation peak of the size observed in the RotaBIS data was deduced from the start of the hospital peak. The values for the variables α, λ, γ, and $y_{k}$ in the two equations of the H–S model were adjusted until the model output produced hospitalisation peaks of a size and frequency that matched the observed data.

Replicating the observed biennial peaks in the model, we identified the drivers of their appearance and how those peaks could be reduced by changing their input values. It should be noted that the observed data reported in Figure 2 were annual numbers, while the data presented in this second model were weekly. The latter should sum each year to the observed annual data.

Sensitivity analysis investigated the effect of changing the pool of susceptibles by 10 percentage points by increasing vaccine coverage by 10 percentage points on the number of hospitalisations in the post-uptake period.

Finally, a scenario evaluation assessed whether increasing the vaccine coverage from the first year in the vaccine uptake period affected the hospitalisation peaks during the post-uptake period. This was simulated by reducing the number of susceptibles in the H–S model by increasing vaccine coverage by 15 percentage points, 30 percentage points, or to a maximum coverage rate of 95%.

### 2.6. Statistical Tools

We used @Risk from Palisade to simulate the distributions and STATA software to statistically evaluate the datasets.

## 3. Results

Table 4 shows the observed data from RotaBIS by year and age group. The two models developed needed to be fitted to replicate these data as closely as possible [19].

### 3.1. The Vaccine Uptake Period

#### 3.1.1. Model Fit

Table 5 shows the results of the model fit for the vaccine uptake period (baseline values with adjusted values obtained by running the Solver software). Figure 5 shows the near equivalence between the observed data and the optimal model fit data. The sum of the observed totals for the first eight years post-vaccination was 3314 (Table 4, sum of ‘Total’ for Y1–Y8), with a result of 3311 for the model with a root mean square deviation equal to 0 (Table 6).

To bring the modelled curve to the level of the observed data, we had to include the impact of secondary sources of infection (Table 1, SIA = 35% and SIB = 33%). The latter infected those who were normally protected by the herd effect of the first vaccination, resulting in a 6% decrease in vaccine effect during the observation period or 210 additional hospitalisation events that could have been avoided. As a result, with vaccine coverage of 52% during the peak season in the first year, secondary infections appeared, causing a reduced herd effect in unvaccinated children in the years after the first vaccination year (yellow cells in Table 4). This is plotted in Figure 5, showing the adjusted line that obtained the best model fit with the observed data.

The model output identified the time-point at which the hospitalisation curve levelled off to a plateau, instead of continuing to decrease at the same rate as previous years. This plateau occurred in year 3 after the first two years of vaccination had reduced the bulk of the disease burden. The hospitalisation level at which the plateau appeared was determined by the coverage rate obtained during the peak season in the first year, as explored in the scenario analysis.

#### 3.1.2. Hypothesis Testing

The fact that the model needed to include secondary sources of infection to obtain the best match to the observed data is consistent with our alternative hypothesis (*H_a_*), that secondary sources of infection affect vaccine effect during the vaccine uptake period.

#### 3.1.3. Scenario Analysis

The scenario analysis in which we improved the vaccine coverage in the first year, leading to a higher herd effect and to a reduced secondary infection rate, shifted the hospitalisation curve lower and to the left. We applied an increase in first-year vaccine coverage rate to the model in steps of 15 percentage points (Figure 6). The second step (Max) came close to the maximum vaccine effect (90% vaccine effect, 90% vaccine coverage, and 85% first-year coverage) shown in Figure 2 as ‘Ideal’.

#### 3.1.4. Sensitivity Analysis

Table 7 shows the results of the sensitivity analysis on the variables that determined the four areas that were not fully restricted by the constrained rules of the optimisation model (the indirect vaccine effect). This analysis defined the variables that had the most impact on the course of the curve during the vaccine uptake period. Depending on the first-year vaccine coverage, the secondary sources of infection played a role in pushing the decreasing curve in the opposite direction to the herd effect. Vaccine waning, here simulated from the third year onwards, had little to no impact because the bulk of the disease was already covered by the vaccination before waning could start having an impact. The results of the adjusted waning scenario in the absence of secondary infection sources show that waning must increase by 200% of its current value (Table 4) to reach the same result as the observed data. Table 7 shows the results of a limited adjusted waning, in which waning was increased by 100%. The number of hospitalisations still only reached 3164, well short of the 3314 in the observed data.

### 3.2. The Post-Uptake Period

#### 3.2.1. Model Fit

Table 8 shows the adjusted values that produced the model fit for the hospitalisation peaks in the post-uptake period, with a value for the area under the curve that was equivalent to the 469 hospitalisations in the observed data at year 9 post-vaccination (Table 4).

Figure 7 shows the profile comparison of the susceptibles and the disease events/hospitalisations using the same H–S model for the pre-vaccination period and the post-uptake period. Pre-vaccination, the disease event peaks occurred on an annual basis, while the post-uptake period of the RotaBIS study showed biennial disease peaks. Vaccination had a major impact on reducing the number of susceptibles in the post-uptake period compared with pre-vaccination.

#### 3.2.2. Hypothesis Testing

Figure 8 shows the age distributions in the hospital peaks of the pre-vaccination period compared with the first small hospitalisation peak in the post-uptake period (year 9 post vaccine introduction (Table 4)). This analysis tests the hypothesis that regular biennial peaks may occur due to an age-shift in the distribution, which occurs because unvaccinated age groups (not directly affected by vaccination) become the primary source of infection. The same relative age distribution in the second hospitalisation peak of the post-uptake period in year 11 post-vaccine introduction was observed as in year 9 (data not shown).

The Figure 8 shows the marked age-shift, with a decrease in the percentage of hospitalisations accounted for by the high-transmitter group aged 3–14 months from 59% in the pre-vaccination period to 35% in year 9 post-vaccination ((Chi-square: 74.15; Df: 1; *p* < 0.0001). This is consistent with our alternative hypothesis (*H_a_*). Conversely, the group aged 15–23 months increased in relative size, from 19% pre-vaccination to 35% post-vaccination. Both age-groups together (3 to 23 months) formed an important susceptible population group during the disease peak season in the post-uptake period, causing the hospitalisation peaks every two years. In absolute terms, the peaks were small (*n* = 305) compared with the primary source in the pre-vaccination period (*n* = 828). It therefore takes a longer time (two years) before the susceptible group is sufficiently large during the peak season to induce a hospitalisation peak. This age-redistribution results in the peaks occurring regularly in frequency and in size every two years instead of annually as in the pre-vaccination period. As shown in Figure 9, the older age-group (13–23 months) contributed most to the primary source of infection in the post-uptake period, whereas the younger group (3–12 months) was directly protected by the vaccine and had a smaller effect. The older group was not under the immediate effect of the vaccination, and therefore the size and reproductive number of this group determined the frequency and size of the hospitalisation peaks.

#### 3.2.3. Sensitivity Analysis

Figure 10 shows the results of the sensitivity analysis in the post-uptake period. Reducing the number of unvaccinated susceptible newborns by 10 percentage points during the post-uptake period by increasing the vaccine coverage rate indicated small improvements in hospital reduction in the short-term (Figure 10).

With differential equations, changing one variable input often affects more than one result at the same time. If the number of newborn susceptibles was changed by a higher vaccine coverage, it had an impact on the number of susceptibles in the group, but also on the reproductive number. This reduction in the number of susceptibles extended the time taken to increase the number of susceptibles, which in turn resulted in smaller hospital peaks over time, proportional to the number of susceptibles available to cause the peak (Figure 10).

#### 3.2.4. Scenario Analysis

Figure 11 shows the effect of increasing vaccine coverage in the first year of vaccination on hospitalisations over 10 years (520 weeks) in the post-uptake period, compared with the H–S model fitted to the observed RotaBIS data (‘Observed’).

Decreasing the number of susceptibles by increasing first-year vaccination coverage by 15 percentage points, 30 percentage points and to 95%, concentrated in the group aged 3–12 months, systematically reduced the frequency and height of the peaks in the post-uptake period. Frequency reduced from biennial disease peaks to every four years, every eight years, and every 10 years as the number of susceptibles was reduced. There was also a dramatic reduction in the height of the peaks, resulting from the lack of susceptibles present in the older age-groups, which caused the greater height in the simulation fitted to the observed data. The analysis illustrates that if the vaccination had been introduced at higher coverage from the first year of vaccination, thereby minimising susceptibility in the group aged 13–23 months, the likelihood is that repetitive disease peaks would be much reduced in size and frequency for a considerable time into the post-uptake period.

## 4. Discussion

The mathematical modelling analysis presented here suggests an explanation for the higher than expected plateau and subsequent biennial peaks in rotavirus hospitalisations observed in the RotaBIS study after the introduction of rotavirus vaccination in Belgium. Our findings indicate that the most important determinant of rotavirus vaccination impact is the coverage rate in the first year. Achieving a high vaccine coverage rate as quickly as possible before the first rotavirus disease seasonal peak maximises the impact of vaccination on reducing rotavirus hospitalisations in at-risk children for many years after vaccine introduction. Conversely, increasing vaccine coverage in later years has relatively little effect. Our model findings indicate that secondary sources of infection are consistent with the observed RotaBIS data, while vaccine waning is not a satisfactory explanation. The analysis presented here extends our previous study on the vaccine uptake period [18] by adding a second mathematical model to cover the post-uptake period.

This study could help decision-makers to design vaccine implementation strategies to achieve better rotavirus vaccination outcomes than those obtained in Belgium. It should be noted that the RotaBIS study results were locally interpreted as a vaccination success, with the reduction in disease-specific hospitalisations of around 73% in the at-risk population [38,39]. The small biennial peaks that appeared after a while were considered small side-effects of the vaccination programme, as vaccination cannot be perfect if the coverage is not 100%. However, the analysis of data from the RotaBIS study presented here indicates that the success of the rotavirus vaccination programme in Belgium could have been greater with a different implementation strategy. An earlier start of vaccination prior to the next seasonal disease peak season could have initiated a cascade of additional disease events being avoided. The additional benefit could have been substantial: in the short-term, a better herd effect; in the medium-term, a much lower plateau in hospitalisations after the third year of vaccination; and in the long-term, smaller and less frequent new disease peaks after reaching a new infection equilibrium over time. Comparing the numbers of hospitalisations over 13 years for the ‘Ideal’ regression line shown in Figure 2 (expected impact of vaccination with 90% vaccine effect, 90% vaccine coverage and herd effect of 85% protection of unvaccinated children) with the numbers of hospitalisations observed in the RotaBIS study indicated that around 3200 more hospitalisations could have been avoided over the 13-year period, close to a 20% improvement. These findings are potentially useful for decision-makers in countries considering the introduction of rotavirus vaccination programmes as they indicate that concentrating on obtaining high coverage as quickly as possible in the first year is the single most important factor in maximising the benefit of the vaccination programme.

The analysis also shows the importance of understanding and identifying the primary sources of infection together with the presence of secondary sources. Therefore, the objective of a new vaccination programme against rotavirus should be to maximise the elimination of the primary source of infection in the first year as a necessary condition for subsequent reduction in the effect of secondary sources. Not achieving that goal immediately allows the virus to remain in the at-risk population in the older age group acting as secondary sources of infection, with the appearance of regular small disease peaks over time. These new peaks are conditional on the remaining susceptible group, with their contact network and transmission risks defining the reproductive number, resulting in regular peaks. This new disease pattern, appearing 7–8 years after the vaccine introduction in the RotaBIS data, is little affected by an increase in vaccination coverage (up to an additional 10 percentage points) in the vaccinated group when coverage is already above 80% during the post-uptake period. Our analysis showed that this is because the primary infection source shifted to an older age group when vaccine coverage was not maximised in the first year. This older age group was larger than the increase in the vaccinated group resulting from the 10 percentage point increase in coverage, and as they are too old for vaccination, the vaccine has no immediate impact on it. Data (not yet disclosed) from the period of the lockdown measures taken to reduce the COVID-19 pandemic in Belgium in 2020 indicated that rotavirus hospitalisations were significantly reduced for that year, although a new disease peak would have been expected. Vaccination coverage rates remained high, and the lockdown measures may have effectively been an opportunistic measure to obtain long-term maximum control of rotavirus infection spread and disease events in Belgium. This may indicate that the biennial disease peaks arising from low vaccine coverage in the first year can be subsequently reduced by drastic measures such as lockdown. If the biennial rotavirus peaks do not reappear in the future, we may never know whether the biennial peaks seen in the RotaBIS study would have been maintained over time.

Compared with the previous reporting of the RotaBIS results [19], the following can be concluded. The two-period modelling presented here, together with the introduction of secondary sources of infection and a likely timescale for vaccine waning, have provided a better insight and explanation of the factors affecting rotavirus vaccination impact over time. This new approach could also be helpful when considering the introduction of other vaccination programmes.

Comparison with other countries may also be of interest to identify similarities and differences compared with our results. We were able to collect data from four countries in Europe that initiated a UMV against rotavirus disease during the past decades, although results were not reported or published beyond the first four to six years of their vaccination programme. These countries are Austria (2007) [12,17,40], Finland (2010) [20,41,42], the UK (2013) [21,43,44,45], and Ireland (2018) [46]. We also considered Australia (2007) [23,24] and the USA (2006) [22,47] as they have reported equivalent or longer periods of observation.

Figure 12 shows the results from the four European countries. The comparison was not straightforward because sufficient details of age distribution of rotavirus disease pre- and post-vaccination were not always available in the publications, duration of follow-up differed by country, no systematic pre-vaccination baseline values by age group were reported, no systematic introduction of rotavirus testing was reported, or results were very compressed. Briefly, the method used to analyse the published European data was to determine annual estimates of the reduction in disease-specific hospitalisations for the same at-risk population (aged <5 years) from the point at which the vaccine was introduced. We used relative values to allow comparative assessments between the country data.

Figure 12 shows that during the first six years of vaccine introduction in those countries, no disease event peaks appeared. As the number of years of observation is limited, such peaks may possibly occur later in their vaccination programmes, as seen in Belgium. Finland and the UK achieved the best results, close to an ideal situation (‘Best’ in Figure 12) because they started their vaccination programmes earlier (July–September) compared with Belgium (November) and Ireland (December) and immediately achieved a high vaccine coverage of newborns (above 90%). This is in contrast to the Austrian situation, where reimbursement was also started early (in June 2007), but vaccination coverage progressed gradually over time (from 50% to 85%). Therefore, despite starting the programme at the optimal time, the slow coverage build-up meant that the programme could not achieve the maximum accumulated vaccine coverage during the first peak season post-vaccine introduction. Ireland started their campaign in December 2018 and could not have achieved a substantial vaccination effect in the first year, showing that starting the vaccination campaign close to the next disease peak season may limit the value of the vaccine in the first year. Figure 12 supports the importance of starting in good time with the vaccination and obtaining the maximum vaccine coverage prior to the first disease peak season after vaccine introduction.

Figure 12 indicates that the plateaus in the hospitalisation reduction in Finland and the UK were formed at a lower level compared with Belgium and Austria. Our results suggest that reaching this low level early after vaccine introduction may have avoided the creation of groups with secondary sources of infection in the at-risk population, and therefore, it is likely that these countries may not see small peaks appearing later, as seen in the RotaBIS study. This could be a valuable topic for future research if more data become available from these countries over time.

The published analyses from the USA and Australia were mainly concerned with obtaining the right numbers of rotavirus disease cases with the diagnosis corrected from the databases available. In Australia, the latest epidemiologic study on the rotavirus vaccine effect was mainly on acute gastro-enteritis data, in which for the first five years after vaccine introduction (2007–2012), the peaks by age groups were more attenuated, except for the group aged more than three years old, four years after the introduction of the vaccine [24]. It could be that this particular increase, as the authors suggested, was in an age group not covered by the vaccine because they were too old to be vaccinated [23]. Our evaluation indicates that the Australian data showed a hospital reduction with the vaccination, in line with data reported in Europe during the first 5–6 years, but we could not identify from the publications an overall vaccine effect value for the at-risk group per year and therefore could not investigate whether a comparable plateau formed in year 3 after vaccine introduction.

The USA data, recently reported in 2019, have some other interesting features [22]. First, the data showed peaks appearing in the unvaccinated groups not after a long period of vaccine uptake such as in Belgium, but 3–4 years after vaccine introduction. A major difference between the USA situation and Europe is that in the USA, there are recommendations for vaccination, but that does not mean that everyone will immediately follow the recommendation and obtain vaccination. The models report vaccine coverage rates of 50% at the start, which could be the reason for the early appearance of disease peaks. As explained in the present analysis of the Belgian data, if vaccine coverage is not high from the start, there is potential for early shifts to age groups that act as new primary sources of infection and are no longer exposed to the direct effect of the vaccine. As such, the results in the USA are broadly in line with our model findings. Our analysis suggests that these biennial peaks are likely to persist for a long time and will not soon reach a situation close to an appropriate control of this infection, as has been suggested [47]. Slight improvement in vaccine coverage rate will lead to better results, but in the short-term, the improvements will not be very spectacular. The case of the USA is interesting to demonstrate the potential trap in rotavirus vaccination, that without high initial coverage in the first year, biennial peaks can rapidly develop in the post-uptake period, much sooner than expected.

This evaluation of the Belgian data has some limitations. Over a 15-year period, there could have been changes in disease management among the participating centres, of which we were unaware, which could have affected the number of disease events hospitalised such as less severe cases being hospitalised because beds were available. We also made assumptions that the catchment area was considered the same during the whole observation period for each participating hospital, which can be questioned because there is a known overall decrease in newborns noted over the years in Belgium. On the other hand, the study was simple to implement, and no extra effort was requested from the personnel in each of the participating centres as we only used data that were already collected in the hospital databases. Rotavirus tests are reimbursed in Belgium until the age of two years, which means that the data collected under that age limit could be considered quite reliable. There was a strong incentive among the participating centres to continue the study if regular updates about the results were provided. The post-uptake model did not include an age-structured design and therefore could not replicate the change in age distribution observed in the RotaBIS data. However, the model was helpful in indicating the direction of the movements caused by changes in the values of the variables in the model.

## 5. Conclusions

In summary, our models indicate that the process of reduction in rotavirus-related hospitalisations through vaccination was driven most strongly by selecting the optimal starting date of the vaccination and rapidly achieving high vaccine coverage before the first peak disease season. In this respect, Finland and the UK followed the best approach among the countries for which data are available, although it is unknown whether this was a lucky decision or an intentional choice. The coverage rate of their vaccination programme in children is high in both countries as they have special vaccination programmes in place for young children. Belgium may have recently received a second chance for better control of rotavirus disease in the at-risk population with the non-pharmaceutical intervention measures taken against COVID-19, in addition to maintaining a high rotavirus vaccine coverage. The USA is an interesting example of how vaccination can deviate from optimal implementation and the potential difficulty of re-adjusting once a new infection equilibrium has been reached if vaccine coverage is not high enough at the start. Rotavirus infection and its vaccine differ from other vaccine-preventable infectious diseases in children in important ways. The disease burden is concentrated in a very small age group, very young children aged up to two years; the infection is highly contagious (R_n_ ≈ >4); the seasonality of the infection is established but its mechanism remains a mystery; and the vaccine has a limited age-indication (2 to <8 months old). These features need to be considered to construct and organise a vaccination programme to obtain optimal disease control from the start to the long-term. To better understand the results of a vaccination programme, it is critical to initiate at launch an adequate monitoring programme and evaluation scheme, provided that sufficient pre-vaccination data have been assembled. This should not be an expensive extra investment and would greatly help in the assessment of the real value of this vaccination.

## Figures and Tables

**Figure 1 viruses-14-00425-f001:**
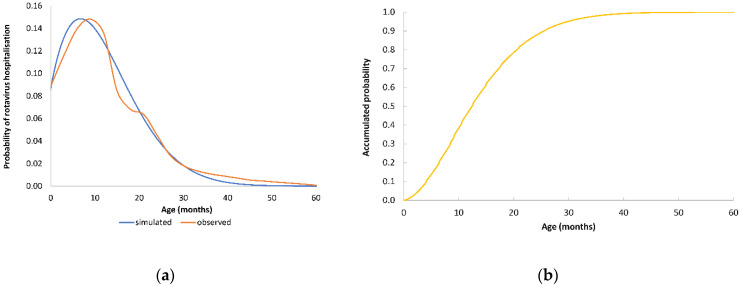
(**a**) The distribution of rotavirus hospitalisations by age in children aged <5 years old in the RotaBIS-study; (**b**) cumulative probability of rotavirus hospitalisation by age.

**Figure 2 viruses-14-00425-f002:**
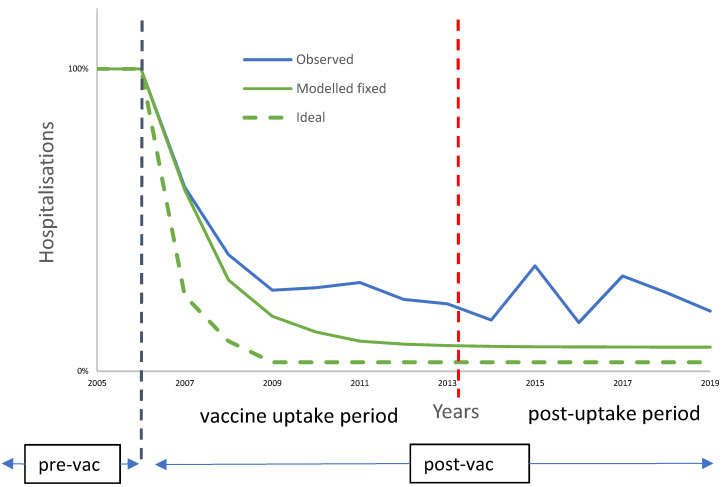
Summary results of the RotaBIS data (‘Observed’) compared with modelled results showing the effect of fixed vaccine effect at 90% and fixed vaccine coverage at 90% (‘Modelled fixed’) and the effect of adding the herd effect at 85% protection of unvaccinated children to the 90% vaccine effect and 90% vaccine coverage (‘Ideal’). The vertical dotted line indicates the two sequential post-vaccination periods modelled separately: the vaccine uptake period in which the coverage is gradually built up after vaccine introduction and the post-uptake period with a new disease/infection equilibrium.

**Figure 3 viruses-14-00425-f003:**
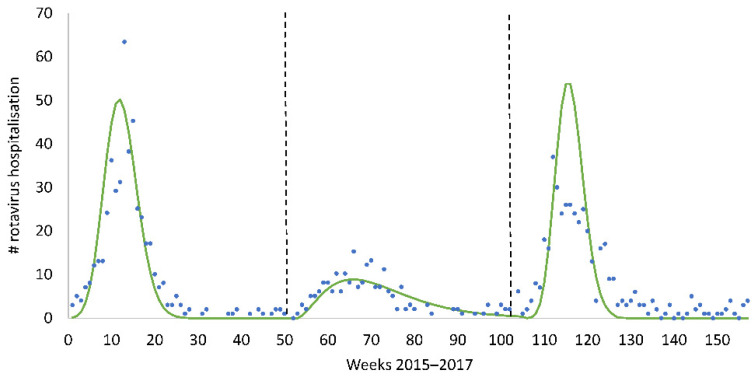
Identifying the biennial hospital peak height in the post-vaccine uptake period.

**Figure 4 viruses-14-00425-f004:**
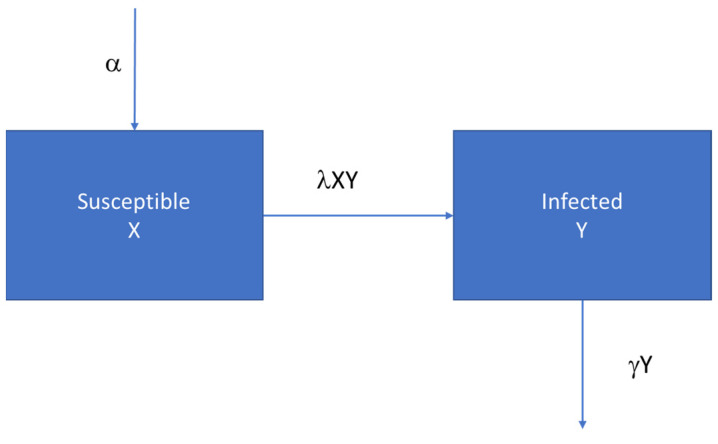
Hamer–Soper (H–S) model design.

**Figure 5 viruses-14-00425-f005:**
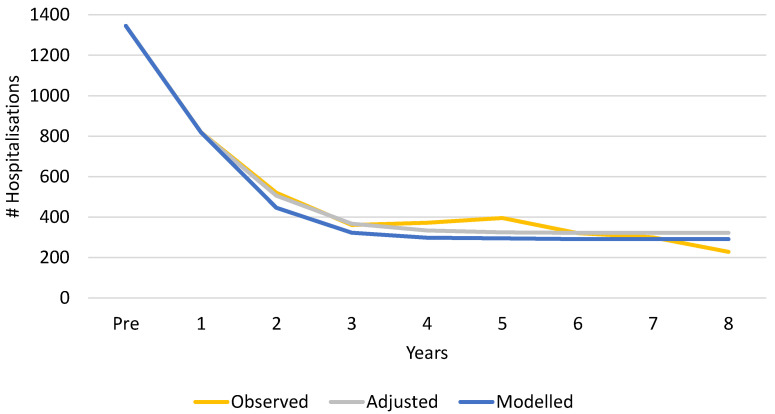
Observed versus modelled and adjusted data for the vaccine uptake period from pre-vaccination to year 8 post-vaccination.

**Figure 6 viruses-14-00425-f006:**
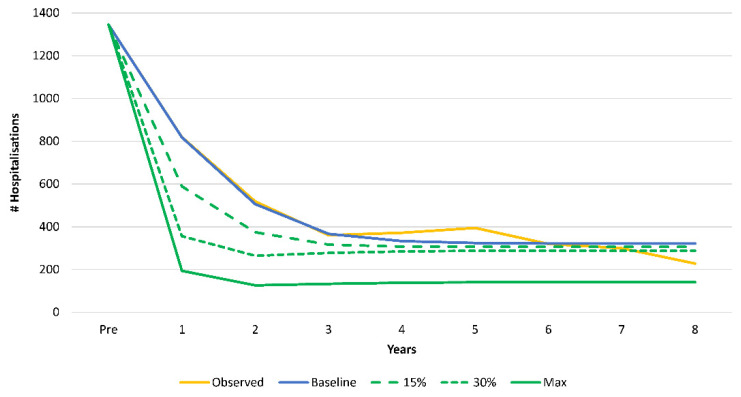
Progressively improving the vaccine coverage in the first year of vaccination resulted in greater herd effect and fewer secondary infections during the first years of vaccination, in turn, reducing hospitalisations.

**Figure 7 viruses-14-00425-f007:**
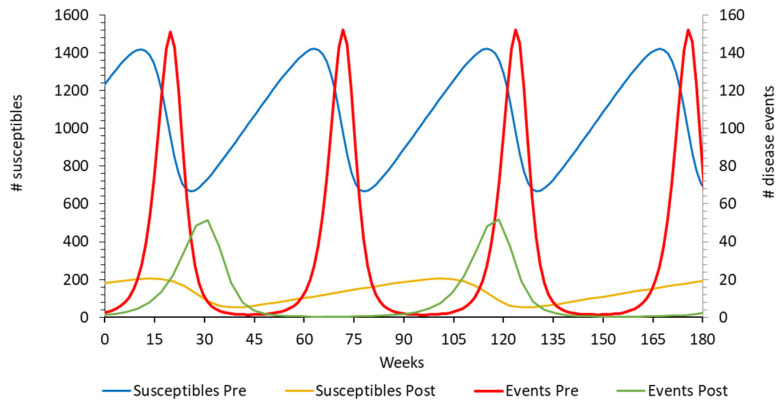
Accrual of disease peaks and susceptibles for the pre-vaccination period and the post-vaccine uptake period.

**Figure 8 viruses-14-00425-f008:**
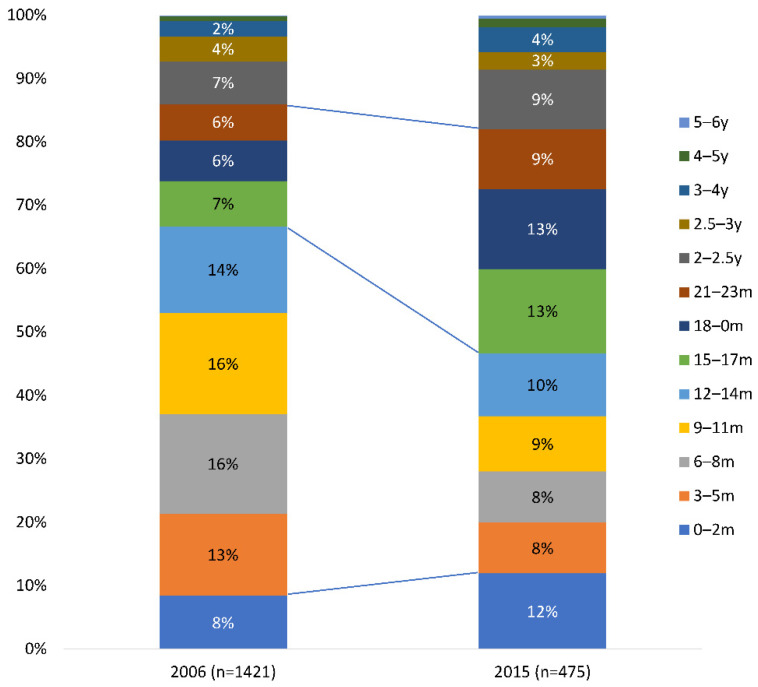
The shift in the age distribution of the disease hospitalisation events during the peak-season from the pre-vaccination period (2006) to the first peak in the post-uptake period (2015). m, month; y, year.

**Figure 9 viruses-14-00425-f009:**
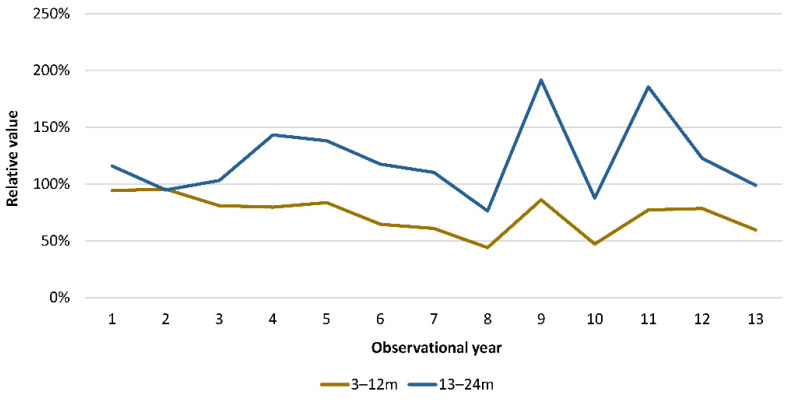
Relative contribution to the deviation of the proportional age-distribution of the vaccine effect by age group per year. m. month.

**Figure 10 viruses-14-00425-f010:**
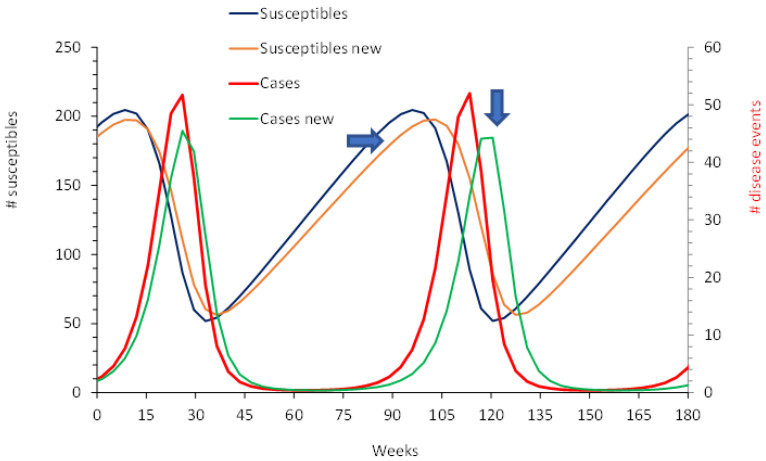
Effect of reducing the number of susceptibles by increasing vaccine coverage by 10 percentage points in the post-uptake period.

**Figure 11 viruses-14-00425-f011:**
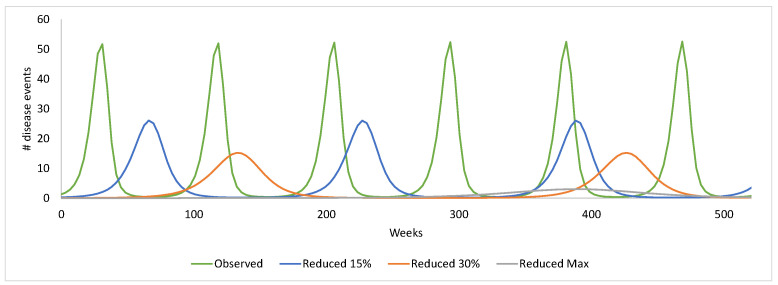
Modelled disease events in the post-uptake period given the condition of hospitalisations in the vaccine uptake period, showing the model fitted to the observed data from RotaBIS, and the effect of reducing susceptibles by increasing first-year vaccine coverage by 15 percentage points, 30 percentage points and to the maximum (95%) modelled (‘Reduced Max’).

**Figure 12 viruses-14-00425-f012:**
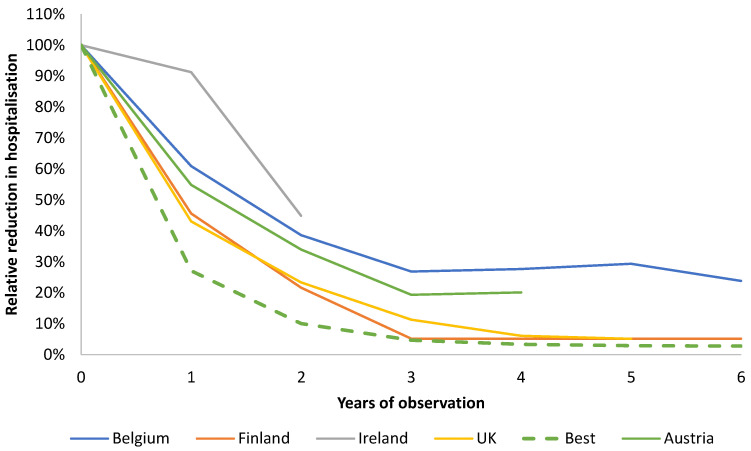
Evaluation of the five countries in Europe that initiated UMV against rotavirus during the past decades. UMV, universal mass vaccination.

**Table 1 viruses-14-00425-t001:** Defining the baseline parameter value estimates from the RotaBIS study (adapted from [18]).

			Uncertainty	
Variable/Force	Code	Value BE	Min	Max
Vaccine efficacy	VE	95%		95%
Vaccine coverage 1st year	Cov1j	52%	49%	54%
Vaccine coverage subsequent years	Covij	83%	82%	85%
Herd effect (0–2 m 1st year)	HEA	15%	13%	17%
Herd effect (older unvaccinated 1st year)	HEB	31%	29%	33%
Herd effect (older unvaccinated subsequent years)	*HED*	*33%*	30%	45%
Herd effect (0–2 m subsequent years)	*HEC*	*75%*	60%	80%
Secondary infection source (2nd year older)	*SIA*	*35%*	30%	45%
Secondary infection source (0–2 m subsequent years)	*SIB*	*27%*	25%	35%
Waning cohort	*Wn*	*12%*	5%	20%

BE, Belgium; m, months.

**Table 2 viruses-14-00425-t002:** Data input for the H–S model simulating the hospitalisation peaks in the post-uptake period.

Variable Name	Code	Value	Minimum	Maximum
Average existing susceptible/wk	$x_{0}$	120	32	180
Existing infectious/diseased/wk	$y_{0}$	1	1	22
Birth rate increase/wk	α	20	5	35
Force of Infection	λ	0.00833	0.00300	0.00900
Time unit (days)	wk	3.5		

Wk, week.

**Table 3 viruses-14-00425-t003:** Defining the areas in the model grid with their regression equations (adapted from [18]).

		A	B	C	D	E	F	G	H	I
	Age Groups	Pre	Year 1	Year 2	Year 3	Year 4	Year 5	Year 6	Year 7	Year 8
1	0–2 months	113	1					2		
2	3–12 months	678								
3	13–24 months	413		5				7		
4	25–36 months	102	3							
5	37–48 months	27				6			8	
6	49–60 months	12			4					
	Total observations	1345								
	Relative	100%								
Subgroup		Area #	Cell numbers			Definition				
Direct vaccine effect		5	B2, C3, D4			First vaccinated birth cohort no waning				
		6	E5, F6			First vaccinated birth cohort with waning				
		7	C2-I2, D3-I3, E4-I4			Subsequent vaccinated birth cohorts no waning				
		8	F5-I5, G6-I6			Subsequent vaccinated birth cohorts with waning				
Indirect vaccine effect		1	B1			Pre-vaccinated period first birth cohort (0 to 2 m) no secondary source of infection				
		2	C1-I1			Pre-vaccinated period subsequent birth cohorts (0 to 2 m) with secondary source of infection				
		3	B2-B6			First year herd effect no secondary source of infection (13 to 60 m)				
		4	C4-C6, D5-D6, E6			Subsequent years herd effect with secondary source of infection (25 to 60 m)				

m, months, Pre, pre-vaccination.

**Table 4 viruses-14-00425-t004:** Observed data from the RotaBIS study by year and age group. Pre-vaccination (blue), vaccine uptake period (green), post-uptake period (orange), and under the influence of secondary sources of infection in the vaccine uptake period (yellow), causing the new peaks in the post-uptake period (brown) (adapted from [18]).

Age Groups	Pre	Y1	Y2	Y3	Y4	Y5	Y6	Y7	Y8	Y9	Y10	Y11	Y12	Y13
0–2 m	113	94	62	56	44	65	54	44	48	56	28	55	52	27
3–12 m	678	340	152	129	127	133	103	97	70	137	75	123	125	95
13–24 m	413	311	208	100	139	134	114	107	74	186	85	180	119	96
25–36 m	102	56	67	49	33	44	33	33	31	67	17	42	37	35
37–48 m	27	16	18	19	19	12	9	15	4	13	8	18	9	9
49–60 m	12	2	12	8	10	7	7	4	1	10	4	6	8	6
Total	1345	819	519	361	372	395	320	300	228	469	217	424	350	268
Relative reduction	100%	61%	39%	27%	28%	29%	24%	22%	17%	35%	16%	32%	26%	20%

m, months; pre, pre-vaccination; y, year.

**Table 5 viruses-14-00425-t005:** Data input in the model simulation.

Variable/Force	Code	Baseline Value	Model Adjusted Value
Vaccine efficacy	VE	95%	95%
Vaccine coverage 1st year	Cov1j	52%	52.6%
Vaccine coverage subsequent years	Covij	83%	82.8%
Herd effect (0–2 m 1st year)	HEA	15%	16.8%
Herd effect (older unvaccinated 1st year)	HEB	31%	30.5%
Herd effect (older unvaccinated subsequent years)	*HED*	*33%*	33.0%
Herd effect (0–2 m subsequent years)	*HEC*	*75%*	78.8%
Secondary infection source (2nd year older)	*SIA*	*35%*	34.8%
Secondary infection source (0–2 m subsequent years)	*SIB*	*27%*	26.5%
Waning cohort	*Wn*	*12%*	18.2%

m, months.

**Table 6 viruses-14-00425-t006:** Sum of results by area for the modelled and observed data and the RMSD.

Area	Modelled	Observed	Difference	RMSD
1	94	94	0.00	0
2	374	373	0.84	−628
3	385	385	0.00	0
4	132	134	−2.03	541
5	597	597	0.00	0
6	23	26	−2.75	136
7	1653	1653	0.00	0
8	52	52	0.46	−48
Sum	3311	3314	−3.48	0.000

RMSD, root mean square deviation.

**Table 7 viruses-14-00425-t007:** Impact of herd, secondary infection, and waning on hospitalisations.

Area	Observed	Modelled	No Secondary Infections	No Herd Effect	Limited Adjusted Waning	No Waning
1	94	94	94	**113**	94	94
2	373	378	**168**	**790**	168	378
3	385	385	385	**554**	385	385
4	134	132	**77**	**192**	**77**	132
5	597	597	597	597	597	597
6	26	23	23	23	**40**	20
7	1653	1653	1653	1653	1653	1653
8	52	52	52	52	**150**	**31**
Total	3314	3314	3050	3976	3164	3289
Difference		0	264	−661	150	25

**Table 8 viruses-14-00425-t008:** Data input adjustment for the best-fit of the H–S model simulating the biennial peaks in the post-uptake period and the data input for the pre-vaccination peaks.

		Post-Uptake Period		Pre-Vaccination
Variable Name	Code	Baseline Value	Adjusted Value	
Average existing susceptible/wk	$x_{0}$	120	120	1020
Average existing disease events/wk	$y_{0}$	1.0	0.4	1.4
Birth rate increase/wk	α	20	19.9	10.4
Force of Infection	λ	0.00833	0.00833	0.00098
Time unit (days)	wk	3.50	3.50	1.24

Wk, week.

## Data Availability

All the data used were retrieved from published papers or documents available on official websites. The models and methodology used in the research were developed by B.S. and are available on request.

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
