# Peer review of "Defining the Recipe for an Optimal Rotavirus Vaccine Introduction in a High-Income Country in Europe"

_viruses, 2022, doi:10.3390/v14020425_

Round 1

Reviewer 1 Report

Summary.

This article studies the data obtained from Belgium's vaccination strategy after 15 years of universal mass vaccination. The authors observed two periods (1 and 2) which showed on period 1 an immediate decrease of the cases of rotavirus, while period 2 was associated with small biennial outbreaks. Furthermore, the authors modeled the data to adjust the infant populations on conditions permitting the success of the vaccination of the strategy. However, even though this study is fascinating, some doubts come to the vaccination implementation in other HIC. Thus:

Suppose the vaccination strategies imposed in UK and Finland were more successful than the result obtained from Belgium. Then, it will be natural to follow the steps provided by these countries, meaning a massive vaccination on susceptible infants (below six months old) months before the winter season (July-September). Therefore, it is unclear why it is necessary to consider the text's proposed considerations.

Another relevant point not mentioned in this study is the possibility of new rotavirus circulating variants among the Belgium population that are not neutralized by the current vaccine. Thus, there is scientific literature regarding new rotavirus strains circulating in Europe that are not neutralized by rotateq and Rotarix.

This manuscript is of common interest but requires to be edited before being considered for publication.

The materials and methods section and the result section are mixed in this manuscript. Many parts of the M&M section correspond to the results. This includes Lines 125-131, 135-141,142-176,301-308, 316-337,364-374, 377-378. This means a reorganization in the result section to re-introduce the above text.

The figures in the manuscript are not well displayed. Thus, it is unnecessary so many single figures but instead, put them together. The following figures should be fused in a unique figure: Figures 3, 4, 6, and 8; Figures 2, 7, and 8; Figures 10 and 11. The authors should consider removing figure 12 of the manuscript and keep figure 13, which should be fused to figure 14.

Author Response

Response to Reviewer 1 Comments:

Comment: This article studies the data obtained from Belgium's vaccination strategy after 15 years of universal mass vaccination. The authors observed two periods (1 and 2) which showed on period 1 an immediate decrease of the cases of rotavirus, while period 2 was associated with small biennial outbreaks. Furthermore, the authors modeled the data to adjust the infant populations on conditions permitting the success of the vaccination of the strategy. However, even though this study is fascinating, some doubts come to the vaccination implementation in other HIC. Thus:

Suppose the vaccination strategies imposed in UK and Finland were more successful than the result obtained from Belgium. Then, it will be natural to follow the steps provided by these countries, meaning a massive vaccination on susceptible infants (below six months old) months before the winter season (July-September). Therefore, it is unclear why it is necessary to consider the text's proposed considerations.

Response: Thank you for the comment. We consider that this research is needed to understand the reasons for the differences in rotavirus vaccination outcomes between countries, and in turn to identify actions that could be taken to improve results in future. When the rotavirus vaccine was first introduced there was no guidance available on optimum implementation, and the fact that different countries selected different strategies indicates that the optimum strategy was not necessarily obvious or natural. Observing different results between countries is interesting, but is of limited value to policymakers without a rationale identifying the factors that drive the differences. Our research indicates that the key feature underlying the greater success of the UK and Finland is likely to be their early start relative to the seasonal peak and high first-year coverage. This is a new finding, and one which we believe will be highly relevant to policymakers designing a rotavirus vaccine implementation programme. It is of particular importance because our research also indicates that sub-optimal early coverage is not easily corrected later and leads to a residual pattern of hospitalisation peaks persisting for many years. Our research also indicates an important role for secondary sources of infection, another new finding as the original dynamic models of rotavirus vaccination did not include secondary sources of infection. Finally, our research indicates that vaccine waning is not a good match for the observed Belgian data, which are better explained by the effect of secondary sources of infection. This is another new finding, and is important because vaccine waning would be key to evaluating potential interventions such as a booster programme or development of new-generation vaccines. We agree that new vaccine-resistant rotavirus strains could potentially appear similar to vaccine waning; however, the resulting pattern of hospitalisations would differ from the observed data (see our response to the next comment).

Comment: Another relevant point not mentioned in this study is the possibility of new rotavirus circulating variants among the Belgium population that are not neutralized by the current vaccine. Thus, there is scientific literature regarding new rotavirus strains circulating in Europe that are not neutralized by rotateq and Rotarix.

Response: We are surprised by this comment because if the reviewer is alluding to the specific strain of G2P4 we cannot say that this strain was dominant in circulation in the RotaBIS study. The European registry on rotavirus strain circulation (the EuroRotaNet study [JID, 2009, 200, Sup1, S215-S221] and the latest EuroRotaNet report of 2019) do not report any dominance of a specific strain so far. Another review publication in 2020 on the dynamic circulation of G2P4 also reported the same conclusions about absence of dominance (Vaccine, 2020.38(35):5591-5600, DOI: 10.1016/j.vaccine.2020.06.059 ). Moreover, if a new dominant strain was circulating, we would not have seen the pattern observed in RotaBIS with biennial peaks. Rather, the models developed indicate that the peaks should increase in height and become annual, similar to the pre-vaccination pattern. We have done this simulation internally, but the graphs were quite different from the observed data. If a new vaccine-resistant strain were present, the youngest age-group should be much more exposed to the virus than reported in Figure 8. We consider that the current observed data are not explained by the presence of a new dominant strain. A new dominant strain could of course appear in the future, but as the observed data do not currently indicate this, we have not discussed it in the paper.

Comment: This manuscript is of common interest but requires to be edited before being considered for publication.

Response: We have extensively reorganized and rewritten the text.

Comment: The materials and methods section and the result section are mixed in this manuscript. Many parts of the M&M section correspond to the results. This includes Lines 125-131, 135-141,142-176,301-308, 316-337,364-374, 377-378. This means a reorganization in the result section to re-introduce the above text.

Response: We were not able to relate the line numbers in the reviewer’s comment to the line numbers in the downloaded document. However, we have extensively reorganized the Material & Methods and Results sections.

Comment: The figures in the manuscript are not well displayed. Thus, it is unnecessary so many single figures but instead, put them together. The following figures should be fused in a unique figure: Figures 3, 4, 6, and 8; Figures 2, 7, and 8; Figures 10 and 11. The authors should consider removing figure 12 of the manuscript and keep figure 13, which should be fused to figure 14.

Response: We have deleted some of the Figures. Some of the fusions proposed were difficult to perform as that would mean that too much information would be presented in some graphs and therefore, we did not proceed.

Reviewer 2 Report

This paper examines the effect of the timing of national rotavirus vaccine introduction relative to the start of the rotavirus season on impact of rotavirus disease.  The hypothesis that timing of national introduction is important and should be chosen strategically is interesting and there appear to be potentially useful findings, but as written, the paper is dense, the sections are mixed together, and the key points are hard to find.  The article feels like a cross between a commentary, modeling study, and literature review.  Some suggestions on how to more clearly present the study and its findings are below.

Major comments:

--Need brief overview of the RotaBIS study as the data from this study seems to be key to analyses but it not described anywhere.

--The methods should consist of overview of data sources, tables of key model inputs and parameters with references (if any of the inputs/parameters are assumptions, the methods to derive them should be provided in a supplemental appendix), hypotheses, model descriptions, and analysis plan.  In the current methods section, all of this information seems to be mixed together despite the section headers.

--The results section is a mixture of results and discussion.  For example, lines 533-536 interpret and discuss the results just presented in the previous sentence.  The fact that this finding is unexpected is more of a discussion point.

--How were the two periods (the vaccine uptake period and the post-uptake period) defined?  Was this based on an increasing vaccine coverage in the uptake period and stable coverage in the post-uptake period?  8 years seems a long time for the vaccine uptake period.

Minor comments:

--Rather than use period 1 and period 2, it is clearer to use vaccine uptake period and post-uptake period throughout.

--Figure 10 – how were the years 2006 and 2015 selected for this comparison?  How representative are they for the other years during the vaccine uptake and post-uptake periods, respectively?  Figure is labeled with calendar years but text describes study years (e.g. 2015 vs. year 9).

--Discussion, line 580:  Consider stating the discussion section with the results from this study (e.g. starting line 580) and then contrasting them with the previously published conclusions from the RotaBIS study (lines 573-580).

--Discussion, lines 626-667 – This is written as a new data analysis section rather than a discussion of the results.  Consider incorporating this into the main methods and results or streamlining the discussion into the main points of comparison rather than this lengthy analysis and comparison.

Author Response

Response to Reviewer 2 Comments

Comment: This paper examines the effect of the timing of national rotavirus vaccine introduction relative to the start of the rotavirus season on impact of rotavirus disease.  The hypothesis that timing of national introduction is important and should be chosen strategically is interesting and there appear to be potentially useful findings, but as written, the paper is dense, the sections are mixed together, and the key points are hard to find.  The article feels like a cross between a commentary, modeling study, and literature review.  Some suggestions on how to more clearly present the study and its findings are below.

Response: Thank you for the comment made. We have tried to follow the suggestions proposed.

Major comments:

-- Comment: Need brief overview of the RotaBIS study as the data from this study seems to be key to analyses but it not described anywhere.

Response: A short explanation about the RotaBIS-study has been added now.

-- Comment: The methods should consist of overview of data sources, tables of key model inputs and parameters with references (if any of the inputs/parameters are assumptions, the methods to derive them should be provided in a supplemental appendix), hypotheses, model descriptions, and analysis plan.  In the current methods section, all of this information seems to be mixed together despite the section headers.

Response: We have followed the suggestion proposed, with some slight variation in the section order because there are two models and two periods to describe.

-- Comment: The results section is a mixture of results and discussion.  For example, lines 533-536 interpret and discuss the results just presented in the previous sentence.  The fact that this finding is unexpected is more of a discussion point.

Response: We have extensively revised the results section.

-- Comment: How were the two periods (the vaccine uptake period and the post-uptake period) defined?  Was this based on an increasing vaccine coverage in the uptake period and stable coverage in the post-uptake period?  8 years seems a long time for the vaccine uptake period.

Response: The separation between the two periods was arbitrary, but based on when the new biennial peaks appeared in the RotaBIS study, the first of which was in year 9. We therefore placed the split at year 8. This is now explained in more detail in the manuscript. We could have used a shorter period such as 6 years instead of 8 for the vaccine uptake period, but it is unlikely to have changed the results seen or the interpretations.

Minor comments:

-- Comment: Rather than use period 1 and period 2, it is clearer to use vaccine uptake period and post-uptake period throughout.

Response: We have made this change across the whole manuscript.

-- Comment: Figure 10 – how were the years 2006 and 2015 selected for this comparison?  How representative are they for the other years during the vaccine uptake and post-uptake periods, respectively?  Figure is labeled with calendar years but text describes study years (e.g. 2015 vs. year 9).

Response: 2006 is the year pre-vaccination in which we knew that the primary source of infection was the children in the age-group 3 to 15 months, which is our reference situation for the primary source in the absence of vaccination. To make a fair comparison with the situation after vaccination, we must select the first new hospitalisation peak after introducing the vaccine, which occurred in 2015, 9 years after introducing the vaccine. We did the same analysis for the second peak of hospitalisation observed in 2017 and the same type of distribution was seen as in 2015. This is now explained in more detail in the text. Our problem was to understand why those small hospitalisation peaks were appearing in a more or less regular pattern every two years. This can be explained if, with the appearance of the secondary source of infection during the vaccine uptake period, they became the primary source during the peaks seen in the post-uptake period. This is the main reason for the selection of the years 2006 and 2015 for making the comparison in Figure 8 in the new manuscript.

-- Comment: Discussion, line 580:  Consider stating the discussion section with the results from this study (e.g. starting line 580) and then contrasting them with the previously published conclusions from the RotaBIS study (lines 573-580).

Response: That is a very good point. We have re-ordered the discussion and added a section on comparison with the previous publication.

-- Comment: Discussion, lines 626-667 – This is written as a new data analysis section rather than a discussion of the results.  Consider incorporating this into the main methods and results or streamlining the discussion into the main points of comparison rather than this lengthy analysis and comparison.

Response: We have reduced the section on comparison with other countries.

Reviewer 3 Report

The study brings clear summary of a 15 years´ observations of rotavirus gastroenteritis incidence after RVA vaccine introduction. The authors did a great work in assessing the optimal model for the maximazing the vaccine´s impact. All the various factors influencing the outcome of the vaccine implementation were taken into account and modelled, then compared with observed data. Even if the topic is more mathematical than virological, the text is comprihensible and nicely readable.

I do not have any suggestions for the manuscript improvement, just some remarks. The main results of the study (it is good to start vaccination as soon as possible after the end of RVGE peak season and achieve high vaccine coverage quickly) are not surprising and could have been predicted. However, the numbers presented in the figures and tables speak clearly and could be used as an argument in countries which are still just considering the rotavirus vaccine introduction. In this sense, I belive that this study could be beneficial. 

Author Response

Response to Reviewer 3 Comments

Comment: The study brings clear summary of a 15 years´ observations of rotavirus gastroenteritis incidence after RVA vaccine introduction. The authors did a great work in assessing the optimal model for the maximazing the vaccine´s impact. All the various factors influencing the outcome of the vaccine implementation were taken into account and modelled, then compared with observed data. Even if the topic is more mathematical than virological, the text is comprihensible and nicely readable.

I do not have any suggestions for the manuscript improvement, just some remarks. The main results of the study (it is good to start vaccination as soon as possible after the end of RVGE peak season and achieve high vaccine coverage quickly) are not surprising and could have been predicted. However, the numbers presented in the figures and tables speak clearly and could be used as an argument in countries which are still just considering the rotavirus vaccine introduction. In this sense, I belive that this study could be beneficial. 

Response: We thank the reviewer for the nice comments.

This manuscript is a resubmission of an earlier submission. The following is a list of the peer review reports and author responses from that submission.

Round 1

Reviewer 1 Report

Towards a recipe of optimal rotavirus vaccine introduction in high income countries

By Baudouin Standaert and Bernd Benninghoff

Submitted to Viruses (Editorial No. viruses-1284849)

General Comments

In this study the impact of rotavirus (RV) vaccines by universal mass vaccination (UMV) on hospitalization for RV-associated acute gastroenteritis (AGE) of children <5 y of age is compared for three European countries – Belgium, Finland, United Kingdom – for which annual post-vaccination hospitalization rates were available – for 13, 5, and 5 years, respectively. The results were compared with an ‘ideal modelled introduction’ (line 85) or the ‘construction of an ideal scenario of vaccine impact’ (line 161), the basic parameters of which are mentioned in Tables 2 and 3B, but the underlying equations of which are never explained nor properly assessed. The RotaBIS data (refs. 27, 28) are cited but incompletely reviewed, and apparently the evaluation has changed several times over the 13 y observation period (Cover letter of senior author to Guest Editor of 15 June 2021).

In detail (see Specific Comments below), various components of the data processing and modelling procedures remain unclear. The Discussion is much too long. Several factors affecting vaccine efficacy/effectiveness, such as the animal RV reservoir or general health factors in low income countries (LIC) are not fully considered. The rationale for the conclusion is not obvious.

Specific Comments

Line

92        … deviations of the curve fit… This remains unclear.

98        … secondary sources of infection cause peculiarities in bad curve setting… Explain what is meant by ‘bad curve setting’.

106ff   Highlights 1 and 2 are too general, highlight 3 as a consequence remains unclear, highlight 4 is not rationally derived and unclear as well.

141      Consider phrasing: … (UMV) campaigns…

146      … expectations made through modelling exercises… Please clarify.

152      The rational of this conclusion remains assumptive.

161      See General Comments.

171      Fig. 1. is not calibrated and superfluous (see Fig. 2, line 232)

180      Table 1. Explain RVGE. The heading ‘Age distribution… etc’ should be moved to above the left hand column.

183/4   Table 2. Variables in the model set up. The latter is not explained.

183      The equations of the modelling construct… Those should be shown and explained.

200      and Table 3. … equations used… ?? Please clarify. Areas and equations are unclear. Table 3B. The ‘equations in each cell’ are components of equations rather than more complex equations used.

223      Fig. 2. … enclosure of the ideal situation… How was this curve obtained? By which mathematical equation is it defined and on which prerequisites is it based? The parts of the observed curves earmarked by arrows and circles should be explained and assessed in more detail.

243      Table 4 describes the data used, but not the function of the model simulation.

247f     Relevant refs of herd effects after RV vaccination should be cited and assessed in context. Consider: Pollard SL, Malpica-Llanos T, Friberg IK, Fischer-Walker C, Ashraf S, Walker N. Estimating the herd immunity effect of rotavirus vaccine. Vaccine. 2015 Jul 31;33(32):3795-800.

252f     Secondary sources of infection. Those should be identified. Have the rich animal reservoirs for species A RVs been considered? There is plenty of evidence for zoonotic transmission of RVAs. Consider providing relevant refs.

259      … small biannual disease peaks… Provide refs. Does the comment relate to Fig. 2? Why should there only be peaks in y9 and y11, and not earlier? Regarding waning immunity consider: Lopman BA, Pitzer VE. Waxing understanding of waning immunity. J Infect Dis. 2018 Mar 5;217(6):851-853.

286      Fig. 3. The generation of the curves shown remains obscure.

292ff   The Discussion is much too long.

307      … catch-up strategy… Please clarify.

317ff   … our modelling approach… to line 342. This text is difficult to follow.

385      … other factors … in developing countries… Consider citation of:

Desselberger U. Differences of rotavirus vaccine effectiveness by country: Likely causes and contributing factors. Pathogens. 2017 Dec 12;6(4):65.

Parker EP, Ramani S, Lopman BA, Church JA, Iturriza-Gómara M, Prendergast AJ, Grassly NC. Causes of impaired oral vaccine efficacy in developing countries. Future Microbiol. 2018 Jan;13(1):97-118.

Reviewer 2 Report

Review: viruses-1284849-peer-review 1

Title:  Towards a recipe of optimal vaccine introduction in high-income countries.

Authors: Standaert B and Benninghoff B

This research article is optimized analysis of Belgium’s experience towards rotavirus vaccination implementation during a period of fifteen years. The analysis was in addition compared to the vaccination programs from Finland and Great Britain. The novelty of this study is based on the introduction of their analysis of two additional factors: secondary sources of infection induced by insufficient vaccine coverage and the age limit for vaccination because of safety issues.  This article is of common interest for the scientific and non-scientific community but requires editing. Furthermore, this article is not written in academic standard language.

Comments.

This article is not written in academic standard language. Therefore, the authors need to write the whole article considering the scientific style for a research article.

Several terms need to be defined earlier in the text as secondary sources of infections. For example, the concept of “Cannibalising the herd effect” is new and somehow contradictory; perhaps the authors should change for more appropriate terms.

Lines 158-176: The methods are not well described and need to be improved to allow the reader to reproduce results. In this sense, a model has been introduced that includes new variables, and therefore, the formula needs to be indicated.  Also, which are the conditions providing “adequate value estimates”? Can the authors provide the number of samples? Are the data only obtained from scientific literature or obtained from government registries? Finally, details about the worst-case scenario should be indicated.

The paragraph from lines 170-177 is unclear. Therefore, a suitable description of this paragraph is necessary for a correct explanation of figure 1.

Table 1 needs editing. Do the authors mean the UK or Great Britain? Since the UK includes England, Scotland, Wales, and Northern Ireland. Great Britain is the largest island of the British Isles. The number of newborns per year in Belgium is unclear. According to https://www.statista.com/statistics/516830/number-of-births-in-belgium/ in 2008 and 2019 were born in Belgium 128’049 and 115’565 infants, respectively. This value is not the number provided in the table.  Can you please explain? Are data available regarding the occurrence of intussusception in Belgium related to the RV vaccine? If Yes, are data correlating age with intussusception?

Can you explain the origin of the data provided in the middle and bottom parts of Table 2?

Values observed in table 3A correspond to modeled or actual values? What is the meaning of the above letters and left-sided letters in tables 3A and 3B? Can you explain the mathematic logic behind the formulas described in Table 3B

Line 220. Change: “The results” to RESULTS

Lines 272-284 are not consistent with Table 1 and Figure 2. Here, it is stated that:” FIN and GB started their vaccination campaign much earlier than B and with a high vaccine coverage rate from start which resulted…”.

The discussion is hard to read and needs to be clarified. Particularly between lines 299-302, 307-310, 371-322, 335-340.